# Topology Preserving Regularization for Independent Training of Inter-operable Models

**Nicolas Zilberstein**[*]
Rice University
nzilberstein@rice.edu

**Akshay Malhotra**
InterDigital Communications Inc.
akshay.malhotra@interdigital.com

**Shahab Hamidi-Rad**
InterDigital Communications Inc.
Shahab.Hamidi-Rad@interDigital.com

**Yugeswar Deenoo**
InterDigital Communications Inc.
yugeswar.deenoo@interDigital.com

## Abstract

Developing schemes to enable zero-shot stitching between different neural networks with minimal or no information exchange has become increasingly important in the era of large and powerful pre-trained models. Considering the example of an autoencoder based data compression framework, having the ability to select the architecture and train an encoder model completely independently of the decoder model while ensuring interoperability between them can revolutionize how these models are developed, deployed, and maintained. In this work, we propose a novel approach that utilizes topological regularizations to align the latent spaces of two different autoencoder models that can be trained independently, without coordination. Our solution introduces two distinct training schemes: *Data2Latent* and *Latent2Latent*. The *Data2Latent* scheme focuses on preserving the topological structure of the input data in the latent space, while the *Latent2Latent* scheme preserves the latent space of a pre-trained, unconstrained model. Through numerical experiments in reconstruction tasks, we demonstrate that our approach yields a near-optimal solution, closely approximating the performance of an end-to-end model.

## 1 Introduction

Compressing data into meaningful low-dimensional representations is a long-standing challenge with applications in image compression [2, 20], audio compression [9], wireless communication [11, 24], and more. Neural networks address this by training an encoder-decoder pair, where the intermediate representation serves as compressed data. The decoder then reconstructs the original data, and the pair is trained end-to-end.

Recent works have studied the *zero-shot stitching* problem [8] in the context of autoencoders, which involves interconnecting encoders and decoders trained independently using limited datasets. In [22], the authors proposed a method that maps the latent representation of each autoencoder to a relative space. However, this approach requires a specialized decoder trained in the relative representation, which is a significant drawback as it necessitates a large amount of data. More recent methods [14, 19] suggest that a linear transformation might suffice to align latent spaces. However, a significant performance gap remains between end-to-end autoencoders and interconnected models when trained on finite, small datasets, suggesting that linear transformations alone are insufficient.

---

[*]Work done during an internship at InterDigital

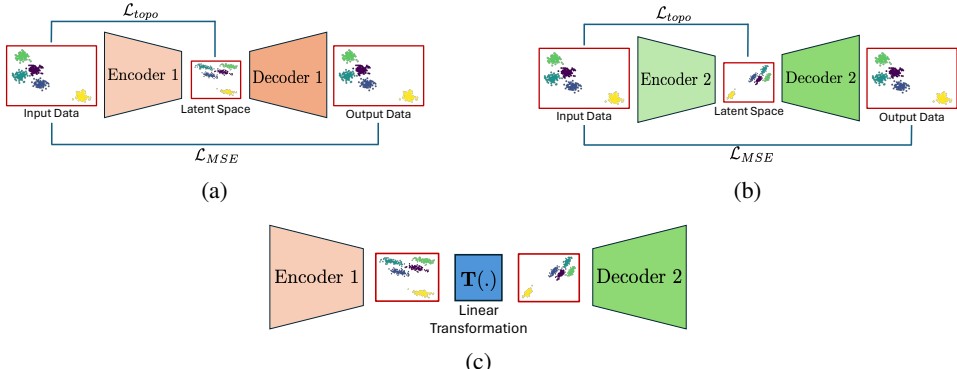

Figure 1: **Data2latent scheme**: a,b) *Topological autoencoders.* In this case, we regularize the training of each autoencoder $AE_1$, $AE_2$ with the input data, preserving its topological structure in the latent space independently of the architecture of the encoder. As an example, we consider a 2D synthethic dataset with 5 classes, its corresponding latent space when considering a bottleneck of dimension two and the topological loss, and the output data which has the same shape as the input. Notice that each autoencoder maps the input data to a different latent space, but both latent spaces preserves the topology up to a rotation and a stretching. c) *Stitching between two regularized autoencoders.* The transformation is simplified, leading to a linear transformation (rotation).

We hypothesize that this performance gap arises due to a lack of geometric similarity between the latent spaces. The encoder's nonlinear mapping from input data to the latent space is often biased by the network architecture, training parameters and weight initializations, making the latent spaces of independently trained autoencoders incompatible when using simple linear transformations.

To address this, we propose preserving the topological features of the input data within the latent space. This regularization encourages architecture-agnostic latent spaces, making them easier to align. Our approach quantitatively measures the dataset's underlying topology and employs it as a regularizer in the training of interoperable autoencoder models. In particular, we build upon the topological regularizer proposed in [21]. Specifically, this regularization aligns the minimum spanning trees of the source (input data) and the target (latent space), leveraging persistent homology to preserve topological features[2]. We propose two training frameworks termed *Data2Latent* (Fig. 1) and *Latent2Latent* (Fig. 2).

**Contributions.** The contributions of this paper are twofold:
1) We propose two training frameworks termed *Data2Latent* and *Latent2Latent*. Each one exploits the topological regularization differently: while the former aims to preserve the structure of the input data, the second one aims to preserve the topological structure of the latent space of unconstrainedly trained autoencoder.
2) Through numerical experiments we show that incorporating topological information facilitates the zero-shot stitching operation with a simple linear transformation.

## 2    Alignment with topological constraints

**Direct alignment between latent spaces.**    Given two autoencoders $AE_1$ and $AE_2$ with latent spaces $\mathbf{Z}_1$ and $\mathbf{Z}_2$ respectively, we seek a method to align their latent space *with minimal interaction* between the models; the ultimate goal is to interconnect the encoder $\mathcal{E}(.)$ of $AE_1$ ($AE_2$) with the decoder $\mathcal{D}(.)$ of $AE_2$ ($AE_1$). Based on the assumption that latent spaces of models trained independently tend to be similar, the authors in [14, 19] propose to estimate a transformation $\mathbf{T}(.)$ by minimizing the mean square error as follow [3]

$$\mathbf{T}(.) = \underset{\mathbf{T}(.)\in\mathcal{T}}{\operatorname{argmin}}\|\mathbf{Z}_1 - \mathbf{T}(\mathbf{Z}_2)\|^2 \tag{1}$$

Although these works claimed that a linear transformation is enough to align the latent spaces of independently trained autoencoders $\mathbf{Z}_1$ and $\mathbf{Z}_2$, the gap in terms of reconstruction error between

---

[2]The referenced paper aligns persistence diagrams obtained via persistent homology, focusing on 0-dimensional topological features, effectively aligning the minimum spanning trees

[3]Although here we focus on the euclidean distance, a different function $\mathcal{L}(.)$ can be used.

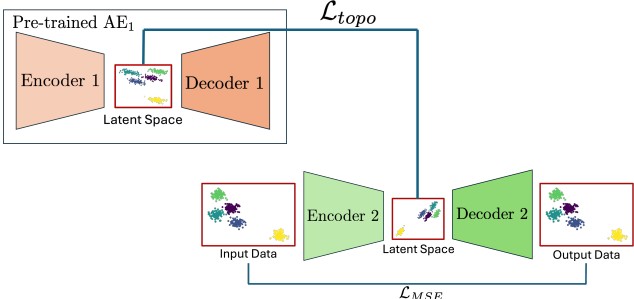

Figure 2: **Latent2latent scheme**. In this case, we regularize the training of second autoencoder $AE_2$ with the latent space of *pre-trained*, *unconstrained* autoencoder $AE_1$. We compute the topological loss using a pre-defined set $\mathcal{D}_{topo}$.

the end-to-end autoencoder and the interconnection is still significant. This difference highlights that the linear assumption for interconnection is insufficient. While a non-linear transformation could potentially improve the alignment [17], it requires a large number of samples from $\mathbf{Z}_1$ and $\mathbf{Z}_2$, violating our goal of minimizing interaction between the models. To cope with this challenge, we propose to incorporate a geometric regularization to *linearize* the relationship.

**Topological regularizations aid similar latent spaces.** We hypothesize that the relationship between the latent spaces can be linear, but achieving this requires an additional constraint. Specifically, we facilitate alignment by incorporating a geometric regularizer that enforces similar topology between both latent spaces. In particular, we aim to make both latent spaces share the same topological features. A natural way to achieve this is by preserving the *shape* of the data in each latent space, which can be done using the topological loss introduced in [21]. In essence, this loss function aligns the minimum spanning tree between a source space and the learnable space by leveraging algebraic topology tools, particularly persistent homology; see Appendices B and C for more details on persistent homology and the topological loss. We incorporate this topological regularization and propose two training schemes that constrain the latent space:

1) *Data2latent*: In this first strategy, we regularize the latent space of both autoencoders so that each latent space retains the same connectivity and topological information as the input space. Formally, we define the loss function as

$$\mathcal{L}_{D2L} = \|\mathbf{x} - \mathcal{D}(\mathcal{E}(\mathbf{x}))\|^2 + \lambda\mathcal{L}_{topo}(\mathbf{x}, \mathcal{E}(\mathbf{x})) \tag{2}$$

where $\mathcal{L}_{topo}(.)$ is defined in (4) and $\lambda$ is a regularization constant. By doing this, and assuming both autoencoders are trained on the same dataset, we constrain the latent spaces to share the same topological structure as the input space, thereby inducing similarity between them. As a consequence, the transformation between $\mathbf{Z}_1$ and $\mathbf{Z}_2$ becomes simpler.

2) *Latent2latent*: In this second strategy, we assume an autoencoder $AE_1$ has been trained without any constraints. Then, given a subset of the training data, $\mathcal{D}_{\text{topo}}$ and the corresponding encoded vectors $\{\mathcal{E}_1(\mathbf{x})|\mathbf{x} \in \mathcal{D}_{\text{topo}}\}$, we train $AE_2$ to minimize the following loss function

$$\mathcal{L}_{L2L} = \|\mathbf{x} - \mathcal{D}_2(\mathcal{E}_2(\mathbf{x}))\|^2 + \lambda\mathcal{L}_{topo}(\mathcal{E}_1(\mathbf{x}), \mathcal{E}_2(\mathbf{x})) \tag{3}$$

where the topological loss is evaluated only on $\mathcal{D}_{\text{topo}}$. Intuitively, during training, we are constraining $\mathbf{Z}_2$ to share the topological structure with $\mathbf{Z}_1$, harmonizing their topological features and simplifying the transformation between them.

Once both models are trained, we estimate the transformation $\mathbf{T}(.)$ by minimizing (1) using a small subset of data points from the dataset denoted as $\mathcal{D}_T$. We consider a linear transformation $\mathbf{T}(\mathbf{Z}_i) = \mathbf{T}\mathbf{Z}_i$.

## 3   Results

We present the results of the proposed scheme on two datasets, MNIST [5] and Fashion MNIST [25]. We utilize two completely different model architectures to show the robustness of the proposed

scheme for the model architectures of the two autoencoders. For the first autoencoder (AE1), a feed-forward architecture is utilized, and for the second autoencoder (AE2), a convolutional neural network architecture is utilized. For MNIST, we consider a bottleneck of 128, while for FashionMNIST 250. Details on the model architecture, hyperparameters, and training can be found in Appendix A.1.

## 3.1 Comparison with other methods

As baselines, we consider the direct alignment with a linear transformation between two unconstrained autoencoders [14] and the relative representation, where a specialized decoder is trained to handle the relative representation [22]. For the direct alignment and our methods, we use 500 samples to estimate the linear transformation, while for the relative representation, we use 1000 anchor points. The normalized mean-square error (NMSE)[4] is shown in Table 1, while qualitatives results are shown in Appendix A.2 in Figs. 4a and 4b. For the quantitative results, we average over five trials with different seeds. The topological constraint harmonizes the latent space, making them easy to stitch: the topological autoencoder achieves an NMSE almost as good as its upper bound ($L_{11}$) for FasionMNIST.

Table 1: NMSE for related and our proposed methods on MNIST and FashionMNIST. $L_{11}$ and $L_{22}$ denote the NMSE of decoding using $AE_1$ (MLP) and $AE_2$ (CNN) respectively, while $L_{21}$ is the NMSE of decoding using directly ($\mathbf{T} = \mathbf{I}$) the encoder from $AE_2$ with the decoder from $AE_1$. We do not include the standard deviation when it is in the fourth decimal place. We consider the average accross 4 trials with different seeds.

| Method | MNIST | FashionMNIST |
|---|---|---|
| $L_{11}$ (MLP-MLP) | 0.020 | 0.018 |
| $L_{22}$ (CNN-CNN) | 0.014 | 0.010 |
| $L_{21}$ (CNN-MLP) | $0.923 \pm 0.002$ | $0.644 \pm 0.061$ |
| Direct alignment (CNN-MLP) [14] | $0.611 \pm 0.081$ | $0.193 \pm 0.03$ |
| Rel. representation [22] | $0.225 \pm 0.003$ | $0.100 \pm 0.003$ |
| Data2Latent (CNN-MLP) (ours) | 0.055 | $0.023 \pm 0.003$ |
| Latent2Latent (CNN-MLP) (ours) | $0.065 \pm 0.001$ | $0.028 \pm 0.002$ |

## 3.2 Ablation

Finally, we do an ablation study of our proposed methods compared to the linear transformation between unconstrained autoencoders [14]. We compare the performance when increasing the size of samples for the linear transformation. For the latent2latent training, we consider $|\mathcal{D}_{topo}| = 20$. The results for $|\mathcal{D}_T| = \{10, 100, 200, 500\}$ and MNIST is shown in Fig. 3a, while for the FashionMNIST is in 3b. We observe that for $|D_T| \geq 200$, the performance of the interconnection between autoencoders gets closer to the upper bound, while the gap between the non-regularized case remains large. Notice that increasing the size of the bottleneck (recall for for FashionMNIST we consider a bottleneck of 250) improves the performance of our methods, achieving a near-optimal performance.

## 4 Conclusion

We study the problem of zero-shot stitching in autoencoders, showing that topological regularization aids a simplified relationship (linear) between two latent spaces. Our proposed solution encompasses two different training schemes: *Data2Latent* and *Latent2Latent*, which aim to preserve the topological structure of the input data and the latent space of an unconstrained autoencoder, respectively. Through numerical experiments on MNIST and FashionMNIST, we showed that our method simplifies dramatically the relationship between two latent spaces. Future work includes studying alternatives geometric losses to align topological features, formalizing the trade-off between performance and samples for the topological loss [12], which is particularly relevant for for the *Latent2Latent* scheme, and expanding the experiments on more complex datasets, such as ImageNet, and of different modalities, such as language and wireless data.

---

[4]We normalize w.r.t. the clean image.

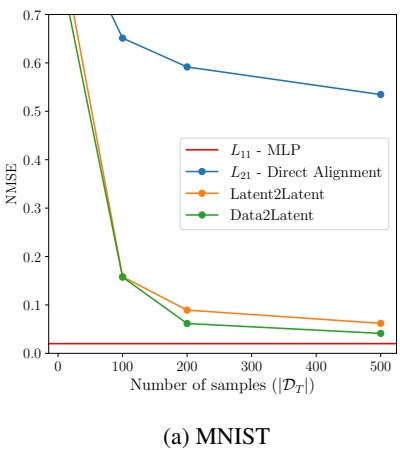
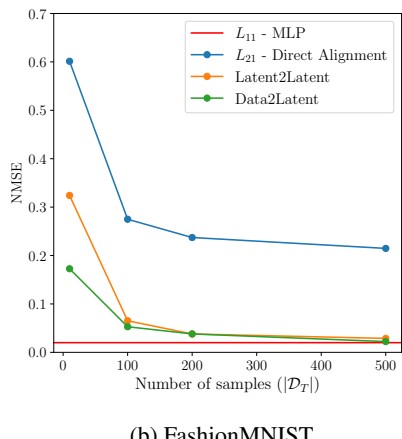

(a) MNIST             (b) FashionMNIST

Figure 3: Performance analysis of our proposed methods as a function of the number of samples ($|\mathcal{D}_T|$) for estimating $\mathbf{T}(.)$ in (1). a) NMSE vs $|\mathcal{D}_T|$ for MNIST. b) NMSE vs $|\mathcal{D}_T|$ for FashionMNIST.

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

## A  Additional details

### A.1  Details of the architectures and training

We now expand on the details of the architectures and training process. The encoder and decoder of $AE_1$ are feed-forward architectures with three layers with ReLU activation function followed by batch normalization and a bottleneck of 128 for MNIST and 250 for FashionMNIST. The hidden dimension at the output of each layer is 1000 for the first layer, 500 for the second one, and 250 for the third. The non-linear activation function at the last layer of the decoder is $\mathrm{Tanh}(.)$. The encoder and decoder of $AE_2$ are convolutional architectures with five convolutional layers. The first layer is a kernel with size of 3 and a stride of 2, with padding to preserve spatial dimensions. The number of channels at the output is 32. The second layer keeps the number of channels constant and has the same kernel size and stride of 1. The third layer has a kernel with size of 3 and a stride of 2, with padding, doubling the number of channels to 64. The next layer preserves the spatial dimensions and retains the same channel depth. The final convolutional layer applies a kernel with size three and a stride of 2 again, maintaining the number of channels. After the final convolutional operation, the output is flattened into a one-dimensional vector, which is then passed through a linear transformation which reduces the feature representation to the same MLP bottleneck (128 for MNIST and 250 for FashionMNIST). Each convolutional layer is followed by a LeakyReLU activation function.

We train for 200 epochs with a batch size of 256. We use ADAM optimizer [13] with a learning rate $l_r = 1 \times 10^{-4}$. For the training scheme *Latent2Latent* we consider $|\mathcal{D}_{topo}| = 20$, i.e., 20 samples

for the topological loss. Lastly, for both schemes we fixed $\lambda = 0.5$ for MNIST and $\lambda = 0.02$ for FashionMNIST. We consider 4 trials per method with different seeds $(0, 100, 1000, 10000)$ and report the average.

### A.2 Additional results

In Figs. 4a and 4b we show qualitative results for MNIST and FasionMNIST for the experiments described in Section 3.1

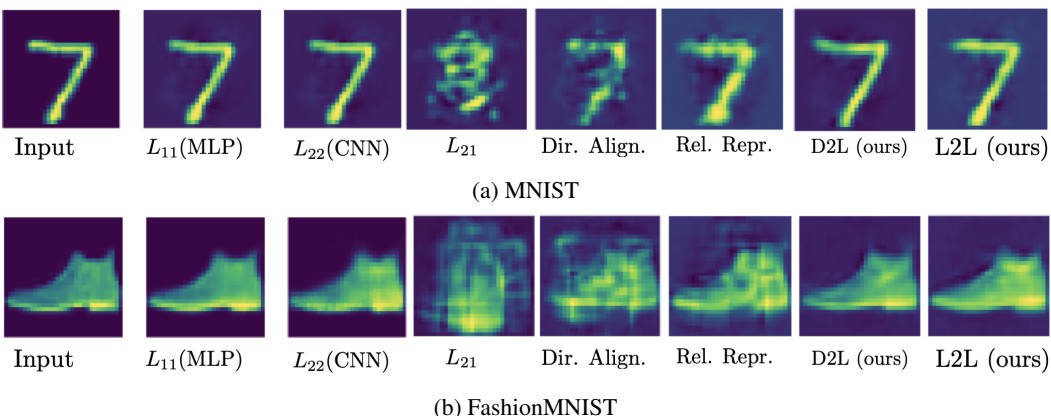

(a) MNIST

(b) FashionMNIST

Figure 4: Qualitative results for all the methods. From left to right: input, $L_{11}$ (MLP), $L_{22}$ (CNN), $L_{21}$ (CNN-MLP), Direct Alignment (CNN-MLP) [14], Rel. Representation [22], Data2Latent (D2L), Latent2Latent (L2). For linear transformation and our two methods, we used $|\mathcal{D}_T| = 500$ samples to estimate the transformation; for Rel. Representation, we used 1000 anchor points.

**Running time.** We compared the training times of the AEs with and without the topological loss. On an NVIDIA DGX A100, the AE without topological loss trains in approximately 10 minutes, while incorporating the topological loss increases the time to around 15 minutes. It's important to note that, in this case, we are only using 0-dimensional persistence features. Future work will include an ablation study on larger datasets to further analyze the impact.

## B Persistence homology

Our method leverages algebraic topology tools [10]. In particular, we build a simplicial complex given a dataset representing a point cloud in a N-dimensional space. A simplicial complex is a set composed of points, lines, triangles, and n-dimensional counterpart objects; for example, a 0-simplicial corresponds to the points of the simplicial. In particular, one-dimensional simplicial complexes are equivalent to an undirected graph. We use persistence homology to find topological features of the dataset, namely connectivity patterns such as connected components; the homology groups describe these topological features.

Formally, we assume that we have a dataset represented as a point cloud, where samples lie in an unknown manifold. Thus, we seek to approximate the unknown manifold. We represent the point cloud as a nested sequence of simplicial complexes; we consider the Vietoris-Rips complex [23]. The sequence is built following a filtration process: for $0 < \epsilon < \infty$, the Vietoris-Rips complex (at each scale $\epsilon$) contains all the simplices of the point cloud whose elements satisfy that $dist(x_i, x_j) < \epsilon \ \ \forall i, j$ [10]. Notice that this construction satisfies the nested relationship, i.e., for $\epsilon_i < \epsilon_j$, the simplicial at scale $\epsilon_i$ is contained in the simplicial at scale $\epsilon_j$; for

The output of the persistence homology calculation of a point cloud are tuples of persistence diagrams $D_d$ and persistence pairings $P_d$, where $d$ is the dimensionality. For example, the persistence pairing $P_0$ contains the edge indices that are in the minimum spanning tree, i.e., the edges that are "relevant" from a topological perspective. The $d$-dimensional persistence diagram contains coordinates $(a, b)$, where $a$ indicates the scale $\epsilon$ at which a $d$-dimensional topological feature is created, while $b$ indicates

the scale $\epsilon'$ at which it is destroyed. On the other hand, the persistence pairing contains indices $(i, j)$ that correspond to simplicies $s_i, s_j$ that create and destroy the topological feature identified by $(a, b)$.

## C  Topological autoencoders

Building a tractable loss function that compares the topological features between the target (input) and learnable spaces is challenging. We consider the loss function defined in [21], which compares the persistence diagrams. We now briefly describe its construction and main ingredients. First, we compute the pair-wise distances between samples in the point cloud (points in the dataset). Let $\mathbf{P}$ be the set of all pairs of data points in the point cloud, the set of all corresponding pairwise distances is denoted as $\mathbf{A}$. We fix the maximum scale $\epsilon = \max \mathbf{A}$ and the maximum dimension $d = 0$. Given this scale, we construct the Vietoris-Rips complex as described above and given the dimension $d$, we keep all the persistence diagrams and pairings up to the dimension $d$. Since, $d = 0$, the persistence diagram corresponds to the minimum spanning tree: recall that the persistence diagram for $0$-dimensional features indicates when two connected components are merged. We do this for both, the target space (i.e. input data space in the Data2Latent setting and latent space of the unconstrained trained autoencoder in the Latent2Latent scheme) and learnable spaces.

Denoting $\mathbf{P_X}$ as the pairing indices of the connected points in the target space (edges of the persistence diagram/ minimum spanning tree in the target space), $\mathbf{P_Z}$ as the pairing indices in the learnable space (edges of the persistence diagram/ minimum spanning tree in the learnable space), $\mathbf{A_X}$ and $\mathbf{A_Z}$ as the set of all possible pairwise distances in the target space and learnable space, respectively. We obtain the distance corresponding to the edges of the minimum spanning trees in the target and learnable spaces as, $\mathbf{D_{XX}} = \mathbf{A_X}[\mathbf{P_X}]$ and $\mathbf{D_{ZZ}} = \mathbf{A_Z}[\mathbf{P_Z}]$, respectively. Finally, we consider the union of all selected edges across the target and learnable spaces, i.e., $\mathbf{D_{ZX}} = \mathbf{A_Z}[\mathbf{P_X}]$ and $\mathbf{D_{XZ}} = \mathbf{A_X}[\mathbf{P_Z}]$. Finally, the topological loss is given by

$$\mathcal{L}_{topo} = \frac{1}{2}\|\mathbf{D_{XX}} - \mathbf{D_{ZX}}\|^2 + \frac{1}{2}\|\mathbf{D_{XZ}} - \mathbf{D_{ZZ}}\|^2. \tag{4}$$

Although we consider the Euclidean norm here, we can also consider other types of distance to compare the persistence diagram. In a nutshell, this loss function combines two terms: the first one seeks to minimize the dissimilarity between the adjacency matrix in the input and latent space when considering the pairing obtained from data graph. Similarly, the second term minimize the difference but when considering the pairing obtained from the latent graph. Considering the union instead of the intersection is essential to have informative gradients; otherwise, if we consider the difference between $\mathbf{D_{XX}}$ and $\mathbf{D_{ZZ}}$, then in the first step, the number of distances selected by the persistence pairing in the learnable spaces would be small due to its random initialization.

## D  Related works

This section expands on the baseline methods and other related works.

### D.1  Relative representation

Relative representation [22] is one of the most popular papers in zero-shot stitching. The work builds on the observation that the representations learned by different neural networks trained on the same data are related via conformal maps; in other words, the angles between latent embeddings are preserved. Based on this observation, they propose to map the latent space of each autoencoder to a pre-defined relative representation, which is invariant by construction to the transformations induced by stochastic factors in the training process. More precisely, they select at random a set of anchor elements $\mathcal{A} \in \mathcal{D}_{train}$; every sample in the anchor set is represented concerning the embedded anchors $\boldsymbol{e}_{\boldsymbol{a}^{(j)}} = \mathcal{E}\left(\boldsymbol{a}^{(j)}\right)$. Then, the relative representation is defined as

$$\boldsymbol{r}_{\boldsymbol{x}^{(i)}} = \left(\mathrm{sim}\left(\boldsymbol{e}_{\boldsymbol{x}^{(i)}}, \boldsymbol{e}_{\boldsymbol{a}^{(1)}}\right), \mathrm{sim}\left(\boldsymbol{e}_{\boldsymbol{x}^{(i)}}, \boldsymbol{e}_{\boldsymbol{a}^{(2)}}\right), \ldots, \mathrm{sim}\left(\boldsymbol{e}_{\boldsymbol{x}^{(i)}}, \boldsymbol{e}_{\boldsymbol{a}^{(|\mathcal{A}|)}}\right)\right) \tag{5}$$

where $\mathrm{sim} : \mathbb{R}^d \times \mathbb{R}^d \to \mathbb{R}$ is a similarity function, yielding a scalar score $r$ between two absolute representations $r = \mathrm{sim}\left(\boldsymbol{e}_{\boldsymbol{x}^{(i)}}, \boldsymbol{e}_{\boldsymbol{x}^{(j)}}\right)$. In particular, they consider the cosine similarity. Notice that the number of anchor points gives the dimensionality of this new latent space; therefore, it is expected

to improve the performance when increasing the number of anchor points. This proposed method requires a specialized decoder trained on the relative representation.

While in the original paper, the authors considered only the cosine similarity, other similarity functions can be incorporated [3, 4]. Lastly, a recent work [7] proposes a method to do zero-shot stitching that leverages functional maps, i.e., by learning a linear transformation in the spectral domain.

## D.2 Autoencoders via geometric regularization

This work focused on topological autoencoders; however, several previous works incorporate a geometric regularization [15]. Many of them build on the notion of *curvature* of a generator [1]. In [16], the authors propose first to build a neighborhood graph on the input data and use it to regularize the geometry and connectivity of the learned manifold. Remarkably, they propose a local approximation of the decoder instead of the encoder to extract local geometric information on the decoded manifold. In [26], a regularization based on conformal mapping, i.e., a mapping that preserves angles, is used to enforce the preservation of angles and relative distances between data and latent space. In [6], inspired by manifold learning, a regularization based on spectral embeddings is incorporated, encouraging the learned latent representation to align with the data geometry. Recently, a graph-preserving autoencoder was proposed in [18], which aims to preserve a pre-build Laplacian between the input data and the latent space.

