# OpenReview forum: "Topology Preserving Regularization for Independent Training of Inter-operable Models"
_NeurIPS.cc/2024/Workshop/UniReps — UniReps_

### Official Review · Reviewer_wkef · 2024-10-03
**Using Topological Loss for Model Stitching**

**Rating:** 7
**Confidence:** 4

**Review:**

**Summary:**

The paper shows that using the "topological loss" from Moor et al. 2020, "Topological Autoencoders" can enable model-stitching of autoencoders through a linear transformation.
This can be both by 1) training two models to match each other where both use a reconstruction loss and a topological loss from input space to latent space
and 2) training a new model to match an unrestricted model by training one model with both a reconstruction loss and a topological loss from the latent space of the pre-trained unrestricted model to the latent space of the model being trained.



**Questions and Suggestions:**

Q1: What effect does the extra loss have on training time?

Q2: You report the "Normalized" Mean Squared Error (MSE). What is the normalization you are using?

Q3: I do not see a report on how many runs you did. Did you only train each model once?

Q4: On line 37, it says: "Our approach quantitatively measures the dataset’s underlying akin topology...". What do you mean by "akin"?

Q5: In the caption of figure 1, the sentence: "At the bottom of each block it is shown an example for a 2D random grid and its
corresponding latent space in 2D." is diffucult to read. I suggest rewriting it.


**Strengths:**

S1: The original work on topological loss, proposes it to make the latent representations more interpretable. Using it instead to enable model stiching is a good idea.

S2: Using the method seems to make the MSE of the stitched model quite close to the original (non-stitched one).

S3: Figure 1 shows the proposed approach nicely.


**Weaknesses:**

W1: From the paper it seems the experiments are only with one run of each model. (I am unsure whether this is a clarity issue or a reproducibility issue.)

W2: I would expect that adding the extra loss would increase training time, but I don't see any mention of it in the paper.
If the training time is not increased by adding the extra loss, this should be mentioned in the paper.


**Justification:**

I recommend this paper to be accepted to the workshop with the added clarifications.

Even if the experiments were only run with one seed for each model, the work still shows that it is possible to stich models trained with the topological loss using a linear transformation.
It would be interesting to dig deeper into the theoretical grounding for this idea.

---

### Official Review · Reviewer_kd4F · 2024-10-07
**Zero-shot stitching using topology-preserving regularization**

**Rating:** 7
**Confidence:** 5

**Review:**

The paper presents a novel solution to the zero-shot stitching problem in neural networks, proposing a topology-preserving regularization method. The methodology is technically sound, leveraging persistent homology to ensure latent spaces maintain topological features, which facilitates stitching across independently trained models.

The numerical experiments on MNIST and FashionMNIST show clear benefits of this approach over previous methods like linear transformations and relative representation. However, these datasets are relatively simple, and the method’s scalability to more complex tasks remains unexplored.

Overall, the approach seems well-executed, but additional results on larger, more challenging datasets (or diverse architectures) would significantly strengthen the claims of the paper. Additionally, while the theory is solid, there are no runtime performance evaluations for the topological loss, which is crucial when considering large-scale models.

---

### Official Review · Reviewer_vjTD · 2024-10-07
**Interesting approach with limited evaluation**

**Rating:** 6
**Confidence:** 2

**Review:**

The paper is about using topological regularization to address the issue of zero-shot stitching. Data2Latent and Latent2Latent training schemes are introduced that seem to offer significant improvements based on the presented results on MNIST and FashionMNIST datasets. However, the evaluation based on these datasets is fairly limited, raising the question about the generalizability of this approach to harder tasks. There is also little comparison with other geometric regularizations. More reasoning on theoretical foundations on why the approach works would also be appreciated.

---

### Author Response · Authors · 2024-10-28

We thanks the reviewer for the constructive feedback. All the comments were important for improving the final manuscript, and future versions of this work. To address the comments/weakness, we added the following to the final version.

1. The details of the training time of each autoencoder. In particular, for these two dataset, the extra loss does not increase the running time significantly.
2. We clarified some things related to the normalization, and the number of runs we did.
3. We corrected typos such as the one in Fig. 1 with respect to the example.

---

### Decision · Program_Chairs · 2024-10-10

**Decision:**

Accept

**Comment:**

In light of the positive reviewers' feedback and relevancy of the submission, we are pleased to accept this paper for presentation at UniReps 2024. We kindly ask the authors to incorporate the reviewers' suggestions and feedback in the final camera-ready version of the manuscript.